# Activity of Ammonium-Oxidizing Bacteria—An Essential Parameter for Model-Based N$_2$O Mitigation Control Strategies for Biofilms

**Arne Freyschmidt *** and **Maike Beier**

Institute of Sanitary Engineering and Waste Management (ISAH), Leibniz University Hannover, Welfengarten 1, 30167 Hannover, Germany; beier@isah.uni-hannover.de
* Correspondence: freyschmidt@isah.uni-hannover.de; Tel.: +49-511-762-2387

**Abstract:** The reduction in N$_2$O emissions is an important task in the control of wastewater treatment plants. Since local operating conditions, especially inside biofilms, are usually not known, models are an important tool in the development and implementation of control strategies. For a pilot-scale nitrifying biofilm reactor and an SBR, different operational strategies to reduce autotrophic nitrous oxide (N$_2$O) formation were developed and tested by applying a combination of modeling and measurement. Both approaches highlighted the relevance of addressing the actual AOB activity as a sensitive control variable. The investigated strategies, therefore, focused on decreasing the AOB-related NH$_4$ conversion rate, as autotrophic N$_2$O formation is directly linked to AOB activity. The results showed that the biofilm system was more advantageous compared with suspended sludge systems. A higher AOB content resulted in a decrease in AOB activity, leading to fewer N$_2$O emissions at the same reactor performance. The highest reduction in autotrophic N$_2$O formation (SBR: 25%; Biofilm: 27%) was obtained by maximizing the aerated time per day and minimizing the number of aeration cycles (the suppression of nitrite-oxidizing bacteria still needed to be ensured). A higher biofilm thickness or a higher sludge mass in the SBR, however, did not have a noteworthy positive effect since no additional biomass could be kept in the system in the long term due to limited substrate availability. Besides nitration, denitrification was also identified as a relevant source of N$_2$O in both systems (biofilm: main source) due to the inhibition of N$_2$O reduction by nitrous acid (elevated nitrite concentrations in combination with pH values < 7).

**Keywords:** greenhouse gas emissions; extended ASM model; biofilm; process control; aeration strategies

## 1. Introduction

The changes currently taking place in the global climate require the wastewater sector to increase its efforts in minimizing the emission of greenhouse gases (GHGs). Besides reducing methane emissions during sludge treatment, a special focus must be placed on biological wastewater treatment. Substantial amounts of the GHG nitrous oxide (N$_2$O) can be formed and emitted during the nitrification/denitrification process if the operating conditions are not adjusted with regard to minimized greenhouse gas formation. Due to high N$_2$O emission factors—up to 20% of the influent nitrogen (e.g., [1])—and the high global warming potential of 265 g CO$_2$e/g N$_2$O [2], the direct N$_2$O emissions often exceed the indirect CO$_2$e emissions caused by energy consumption (e.g., [3]). However, several investigations on large-scale plants have also shown that N$_2$O emissions close to zero are achievable for low-load systems (e.g., [4,5]). In high-load systems (e.g., reject water treatment), emission factors of <0.5% of the influent ammonium can be achieved through optimized operation (e.g., [6]).

N$_2$O is formed by ammonium-oxidizing bacteria (AOB) as a by-product of nitration and by heterotrophic bacteria (HET) as an intermediate of heterotrophic denitrification. For AOB, a strong correlation between N$_2$O formation and the ammonium (NH$_4$) conversion

rate has been reported [7–9]. Moreover, autotrophic $N_2O$ formation is favored at high nitrite ($NO_2$) concentrations [8,10]; the critical $NO_2$ concentration is influenced by adaptation processes (e.g., >20 mg/L for a high-load nitritation system [8] and >3 mg/L for a mainstream nitrification/denitrification system [11]). Above a site-specific limit concentration, no further increase in $N_2O$ formation was detected (e.g., [8]). Oxygen ($O_2$) availability was also identified as a main influencing factor; $O_2$ concentrations of <0.5 mg/L are considered critical ([12–14]). In addition, the $O_2$ concentration influences the activity of HET. Denitrification can act as both an $N_2O$ source and an $N_2O$ sink (e.g., [15,16]) depending strongly on the nitrous acid ($HNO_2$) concentration [16,17]. For this reason, denitrification is a key process in terms of mitigating $N_2O$ emissions.

Currently, activated sludge systems with suspended biomass are predominant, but biofilm systems are also used in wastewater treatment due to their process stability and the potential to use small reactor volumes and a high sludge age at the same time. In biofilms, the $O_2$ concentration varies with depth, allowing different processes to take place simultaneously. Regarding the minimization of $N_2O$ emissions, there is potential for simultaneous denitrification of formed $N_2O$ in the anoxic zones (e.g., [18–21]); however, unavoidable areas with low $O_2$ concentrations inside the biofilm can also favor autotrophic $N_2O$ formation, e.g., [18].

Over the past years, different control strategies and, in some cases, control algorithms for the minimization of $N_2O$ emissions have been published (e.g., [16,22,23]). However, the high number of influencing factors and the variability of local boundary conditions, especially in biofilms, make it difficult to directly transfer control strategies between plants, and site-specific development of control strategies based on measurements is time-consuming and expensive. In this context, the authors propose employing validated mathematical models that can be calibrated with data usually available in practice in combination with short measurement campaigns [23]. Using the models, the different processes of $N_2O$ formation, conversion, and emission can be separated and individually investigated. Subsequently, targeted measures aiming for a balance between maximized reactor performance and minimized greenhouse gas emissions can be developed.

The authors expect that in the future, the use of self-learning artificial intelligence (AI) will also gain importance in the development of operational strategies and plant control. However, in most cases, there are not enough data sets available for AI training, so deterministic models will continue to be needed. By linking deterministic models with methods of AI, existing knowledge can be combined with the strong data analysis capabilities of AI. A major challenge is, however, the automatic evaluation of the quality of the model output, e.g., evaluation based on a statistical evaluation of the specific input data as well as the training data.

In this work, the combined measurement and modeling approach was applied to a sequencing batch reactor (SBR) and a fixed bed biofilm reactor, aiming to minimize autotrophic $N_2O$ formation by reducing the AOB-related $NH_4$ conversion rate. For this purpose, the employed ASM model [23,24] was extended to include the active AOB biomass in the calculation of the $N_2O$ formation factor. The effect of the tested measures on heterotrophic $N_2O$ formation/conversion was also investigated, although improved reduction of formed $N_2O$ by denitrification was not considered, as this topic was already addressed in [20].

In this publication, aerobic conditions were defined as an environment with an oxygen concentration of >0.1 mg/L, and anoxic conditions were defined as an environment with an oxygen concentration of <0.1 mg/L (see Section 2.1).

## 2. Materials and Methods

### 2.1. Definitions

In this publication, the following expressions are used to describe the aeration regime of the pilot plant (intermittent aeration):



- Aeration cycle: summed duration of one aerated phase and one unaerated phase of intermittent aeration (indicated as "time on/time off" in minutes);
- Aerated/unaerated phase: time of one aeration cycle during which the aeration was switched on/off;
- Aerobic/anoxic phase: time of one aeration cycle during which the $O_2$ concentration was $>/<0.1$ mg/L;
- Aerated/unaerated time per day: summed time per day during which the aeration was switched on/off [h/d];
- Aerobic/anoxic time per day: summed time per day during which the $O_2$ concentration was $>/<0.1$ mg/L [h/d].

*2.2. Pilot Plant Description and Measurement*

For this modeling study, the measurement data of a pilot plant were employed. The main characteristics are summarized below:

- Two-step deammonification in two covered biofilm reactors arranged in a series (the biomass content of the bulk phase was negligible);
- Volume per reactor: 220 L working volume, 10 L headspace;
- Textile biofilm carriers (Cleartec® BioCurlz, Jäger, Hannover, Germany) providing a theoretical surface for biofilm growth of 141 $m^2$ per reactor;
- Intermittent aeration in the nitritation reactor (constant air volume flow, $O_2$ concentration in the bulk phase $> 4$ mg/L during the aerated phase) to suppress the activity of nitrite-oxidizing bacteria (NOB); no aeration in the second reactor (Anammox).

In this study, only data from the nitritation reactor were considered since the focus was on investigating the influence of different aeration strategies on $N_2O$ emissions. For comparison, an additional nitritation reactor (220 L working volume) was operated as an SBR with intermittent aeration and suspended biomass. The SBR was continuously fed for 10 h (50 L/operation cycle and 100 L/d) with the same wastewater as the biofilm reactor. The filling phase was followed by a settling phase (1 h) and a decanting phase (1 h).

Both reactors were equipped with online temperature sensors (Endress–Hauser AG, Reinach, Switzerland), online pH sensors (Sensortechnik Meinsberg, Waldheim, Germany), and online $O_2$ sensors (Hamilton, Reno, NV, USA). $NH_4$-N, $NO_2$-N, nitrate–nitrogen ($NO_3$-N), alkalinity, total chemical oxygen demand (COD), and filtered chemical oxygen demand ($COD_{filtered}$) were determined weekly with cuvette tests (Hach Lange GmbH, Düsseldorf, Germany). $N_2O$ concentrations in the reactor (liquid phase) and the off-gas (gas phase) were measured with $N_2O$ microsensors (Unisense A/S, Aarhus, Denmark) several times over periods of multiple days.

For the comparison of the technologies presented here, the operational data of the SBR over a period of 65 days were used as the baseline scenario (see Figure 1). This period was selected because several boundary conditions were fulfilled, allowing the reliable calculation of nitrogen conversion rates and $N_2O$ formation and emission rates:

- Stable nitritation;
- No operational disturbances;
- Long $N_2O$ measurement campaign.

The baseline SBR scenario was characterized by an average $NH_4$-N conversion of 25.5 g/d or 34% of inflowing $NH_4$-N, limited by alkalinity. The converted $NH_4$ was almost completely transformed into $NO_2$ (>99%).

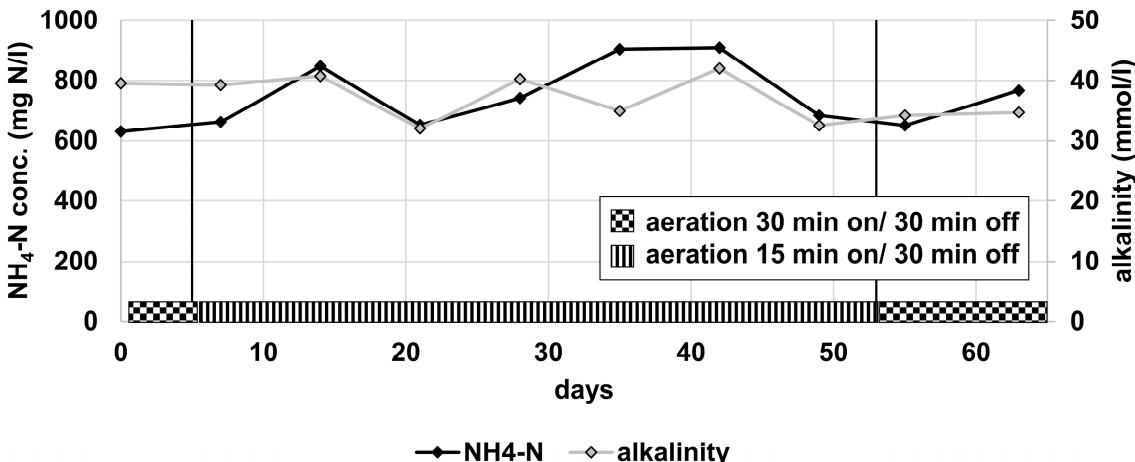

**Figure 1.** Measured influent concentrations of NH$_4$-N and alkalinity as well as aeration mode over the investigated operation period used as input for the simulations.

### 2.3. ASM and Biofilm Model

An extended ASM model developed by [23] and adapted for biofilm modeling by [20] was applied (provided here: [24]). The ASM model includes the following biological and physical processes (see Appendix A):

- Aerobic carbon conversion;
- Two-step nitrification (NH$_4$-N $\rightarrow$ NO$_2$-N $\rightarrow$ NO$_3$-N);
- Autotrophic N$_2$O formation based on N$_2$O formation factors calculated for each time step depending on the NH$_4$-N conversion rate as well as the concentrations of O$_2$ and NO$_2$-N;
- Three-step denitrification (NO$_3$-N $\rightarrow$ NO$_2$-N $\rightarrow$ N$_2$O-N $\rightarrow$ N$_2$-N);
- Anaerobic ammonium oxidation;
- Hydrolysis and ammonification;
- Growth and decay of AOB, NOB, HET, and AMX;
- Aeration, described by O$_2$ transfer rate (g/m$^3$N/m) and air volume flow (m$^3$/h);
- Activation and deactivation of AOB and NOB depending on alternating aerobic and anoxic phases;
- N$_2$O gas transfer and emission.

For biofilm modeling, a one-dimensional approach was selected (the biofilm thickness was calculated dynamically for each simulation step depending on the biomass growth as well as the attachment and detachment rates). The biofilm was divided into four completely mixed layers that interacted with each other through diffusion processes.

During the pilot plant operation, anammox microorganisms started to establish in the biofilm nitritation reactor with ongoing operation, as described in [20]. In this study, however, the anammox activity was set to zero in the simulations to ensure the comparability of both reactors. Otherwise, the NO$_2$ concentration in the biofilm reactor would diminish over time, resulting in lower N$_2$O formation factors and less inhibition of N$_2$O denitrification by HNO$_2$.

### 2.4. Calibration and Investigated Scenarios

The models were calibrated (target parameters: N$_2$O concentration in the bulk phase, N$_2$O emissions, NH$_4$ conversion) using measurement data from the pilot plant operation. For the SBR, data from the investigated 65-day period were employed. The calibration of the biofilm model was described in [20]. Both reactor models were initially simulated with continuous aeration until a steady state (constant biomass concentration) was achieved. Starting from this steady state, the aeration was switched to intermittent ("30/30"), and simulations were performed until a stable ratio between active and inactive autotrophic biomass was established. These states served as the starting point for dynamic modeling.

In the baseline scenarios, the dynamic influent data and operational settings of the investigated 65-day period were applied to both calibrated models to ensure that both reactors were exposed to exactly the same operating conditions, allowing the direct comparison of the $N_2O$ formation and emission processes.

Subsequently, different scenarios were investigated to determine how the specific AOB activity could be reduced while maintaining the same reactor performance (N conversion) in order to diminish $N_2O$ formation by AOB. Considering the unit of the AOB activity (mg N/mg XAOB/d), the following approaches were identified:

1. An increase in the time available for aerobic N conversion by adapting the aeration cycle;
2. An increase in the AOB biomass in the reactor by increasing the biofilm volume or the TSS concentration.
3. A reduction in the N conversion peaks by equalizing the inflow (additional storage tank).

Following these approaches, the scenarios in Table 1 were investigated. In each step, the preferred settings determined in the previous step were used for the unchanged parameters (stepwise optimization).

**Table 1.** Investigated scenarios.

| | |
|---|---|
| S1: different aeration cycles<br>$O_2$ concentration = 4 mg/L | "15/30"<br>"15/15"<br>"30/30"<br>"30/15"<br>Continuous aeration |
| S2: different $O_2$ concentrations<br>SBR: aeration cycle = "30/30"<br>BF: aeration cycle = "15/15" | 2 mg/L<br>3 mg/L<br>4 mg/L |
| S3: increased biomass content<br>$O_2$ concentration = 4 mg/L<br>SBR: aeration cycle = "30/30"<br>BF: aeration cycle = "15/15" | SBR: increased sludge settling abilities<br>Biofilm: decreased erosion velocity |
| S4: equalization of the influent<br>$O_2$ concentration = 4 mg/L<br>BF: aeration cycle = "15/15" | Target value for the inflowing N load (here: overall mean of the total inflowing N load) |

## 3. Extension and Adaptation of the Basic Model

### 3.1. Approach for Calculating an $N_2O$ Formation Factor as a Function of AOB Activity

According to [23], $N_2O$ formation by AOB can be dynamically computed as a fraction of converted $NH_4$, relying on $N_2O$ formation factors calculated by summing the influences of the volumetric $NH_4$-N conversion rate (g N/$m^3$/d) and the concentrations of $NO_2$-N and $O_2$. The volumetric $NH_4$-N conversion rate is used as an indicator of the AOB activity. However, since the specific AOB activity does not depend on the reactor volume but rather the ratio of nitrogen load and converting biomass (g $NH_4$-N/g $AOB_{active}$), the absolute mass of active AOB in the reactor or the biofilm section evaluated must be considered (e.g., [7,8]). It was decided not to select a VSS-related conversion rate because this parameter is mainly influenced by the growth of heterotrophic microorganisms. Instead, a new $N_2O$ formation factor considering the specific AOB activity was introduced to the model on the basis of the following assumptions:

- At an upper threshold for the AOB-related $NH_4$ conversion rate (g $NH_4$-N/g XAOB/d), an $N_2O$ formation factor of 4.74% (determined for high-loaded systems by [16] was applied.
- At a lower threshold for the AOB-related $NH_4$ conversion rate (g $NH_4$-N/g XAOB/d), an $N_2O$ formation factor of 0.74% (determined for low-loaded systems by [16] was applied.

- Between these thresholds, the formation factor was calculated by linear interpolation. In addition, [7] and [8] determined a linear relationship between $N_2O$ formation and bacterial activity for common conversion rates in wastewater treatment.

The $N_2O$ formation factors related to the concentrations of $NO_2$ and $O_2$ do not need to be changed since these parameters are neither variable over the biofilm depth nor dependent on the load situation. The new approach can be used to simulate the effects of a variable AOB activity on $N_2O$ formation and to implement model-based control algorithms to minimize $N_2O$ emissions.

*3.2. Model-Based Determination of the Thresholds for the AOB-Related $NH_4$ Conversion Rate*

Since the thresholds for the AOB-related $NH_4$ conversion rate are difficult to determine by measurement (the AOB mass in the system must be known), an approximation was made by a simplified ASM model for autotrophic metabolism instead.

For the upper threshold, the maximum achievable $NH_4$ conversion rate under the expected site-specific mean boundary conditions (availability of $NH_4$, $O_2$, and alkalinity; inhibition by $NH_3$ and $HNO_2$; and temperature) was applied. This rate was calculated by decreasing the AOB content of the reactor until the minimum AOB biomass required for the desired N conversion was reached. The lower threshold corresponded to the $NH_4$ conversion rate that occurs under standard mainstream wastewater treatment conditions.

According to these results, the relationship depicted in Figure 2 was derived.

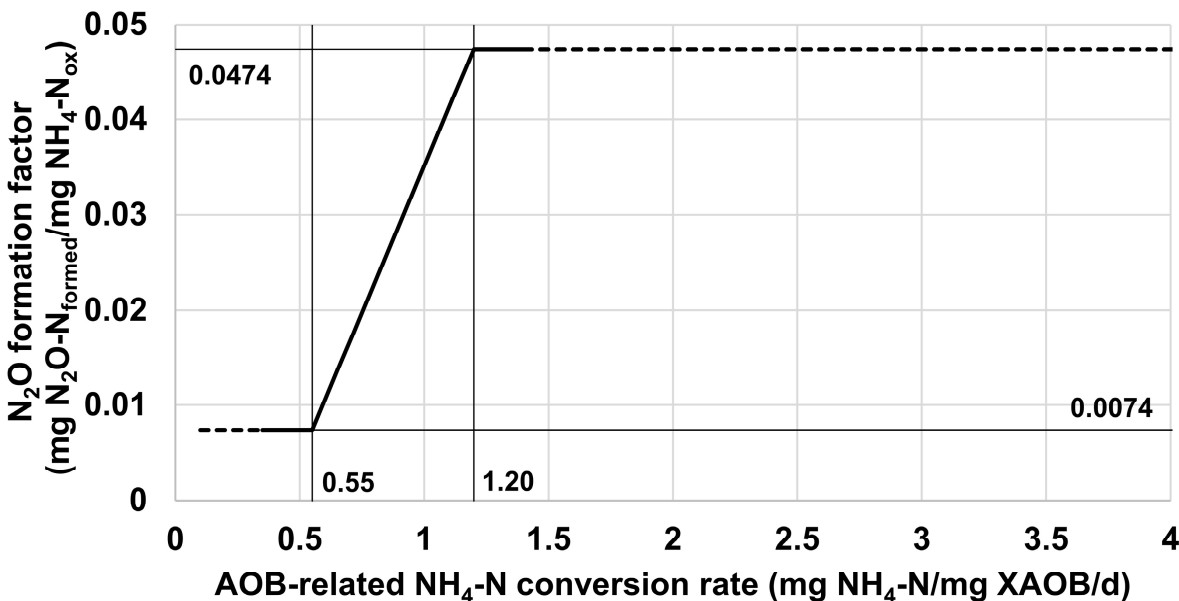

**Figure 2.** Determination of the $N_2O$ formation factor depending on the AOB-related $NH_4$-N conversion rate.

As shown in the next section, the developed approach can be applied to depict the $N_2O$ formation of the investigated pilot plant. For other applications, a site-specific adjustment of at least the upper threshold for the AOB-related $NH_4$ conversion rate is necessary.

The implementation of the formation function in combination with the currently calculated active AOB biomass allows a dynamic description of $N_2O$ formation, e.g., under changing oxygen penetration depths in biofilms. This model extension is a key tool for the design of aeration control in biofilm systems, as investigated in the following section, since the AOB activity directly depends on the local $O_2$ concentration, and increasing or decreasing conversion rates influence the $N_2O$ formation.

## 4. Scenario Analysis and Discussion of the Results Regarding the Optimal Control Strategy

### 4.1. Baseline Scenario SBR

Figure 3 shows the measured and simulated $N_2O$ concentrations in the liquid phase. Due to stripping, a strong decrease in the $N_2O$ concentration was observed in the aerated phase (measurement and simulation), indicating that the stripping rate was higher than the net $N_2O$ formation rate. During the aerated phase, a relatively constant (over the period analyzed) $N_2O$ formation factor of approximately 6.7% of converted $NH_4$ was calculated. A significant reduction in the formation factor (formation factor = 3.3% at the end of the working phase, when the concentration of active AOB biomass was at its highest) could only be observed between day 53 and day 60. In this period, a longer aerated time per day (10 h/d) coincided with a low influent load, resulting in a lower AOB-specific $NH_4$-N conversion rate (see Figure 4) and thus a lower formation factor (see Figure 2).

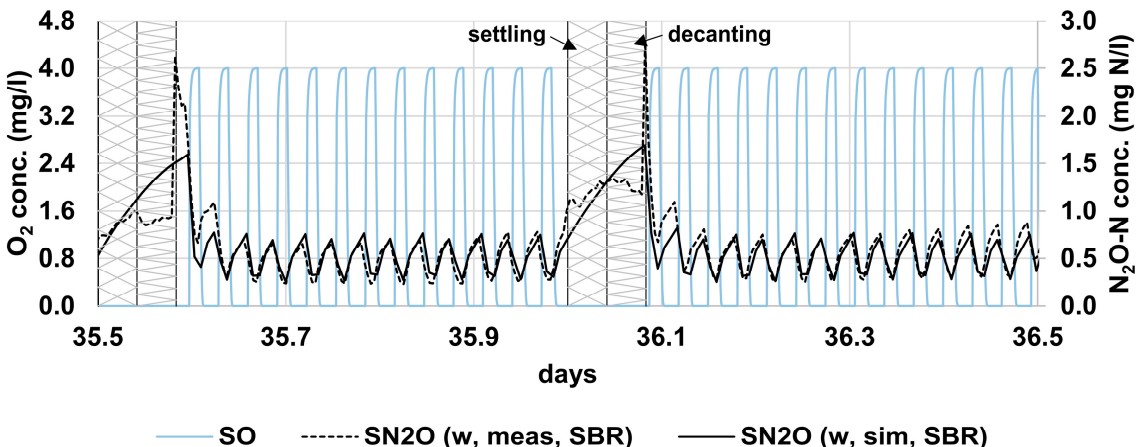

**Figure 3.** $O_2$ profile (simulated) and $N_2O$ concentration profiles (measured and simulated) for the SBR (water phase).

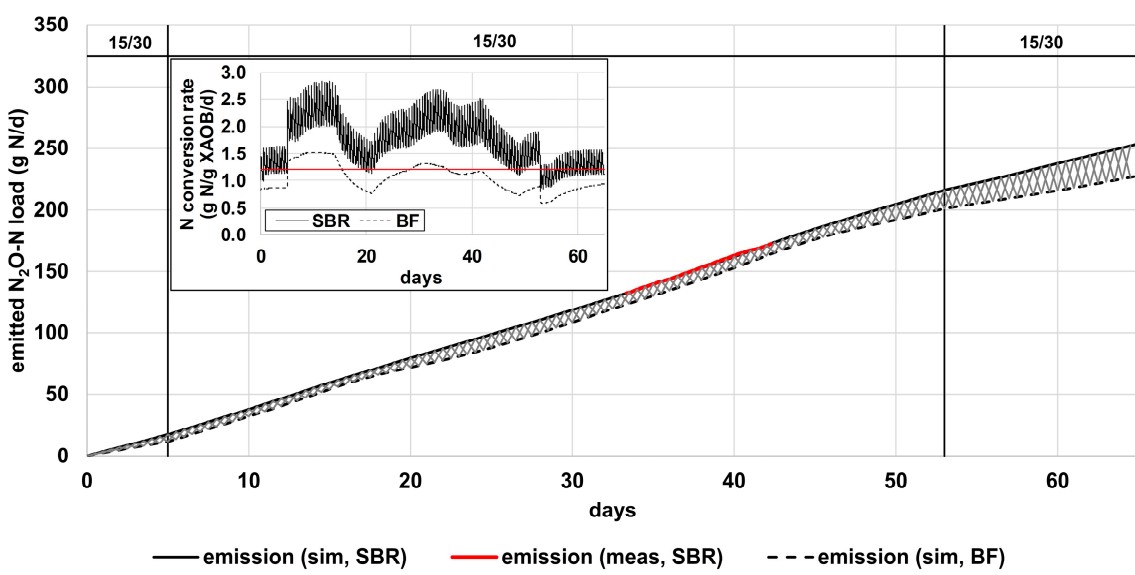

**Figure 4.** Total cumulated $N_2O$ emissions and AOB-related $NH_4$-N conversion rate (red line: critical conversion rate, see Figure 2) of the biofilm reactor (simulated) and the SBR (measured and simulated).

In the unaerated phase, $N_2O$ was initially formed by AOB as long as sufficient $O_2$ was still present (for approximately 5–8 min). At the end of this period, very high formation factors of close to 100% of converted $NH_4$-N were observed due to $O_2$ deficiency. Subse-

quently, a change in the slope (Figure 3) marked the point in time at which the residual $O_2$ was completely consumed (no remaining AOB activity), and $N_2O$ formation was attributed to the activity of HET using the carbon contained in the influent as well as the carbon released during the decay of the biomass. Since $N_2O$ reduction was almost completely inhibited due to the high $HNO_2$ concentrations caused by the high $NO_2$ concentrations and low pH, the denitrified $NO_2$ was only reduced to $N_2O$, but not to $N_2$. Altogether, incomplete denitrification contributed 39.6% to the total $N_2O$ emissions (see Table 2).

**Table 2.** $N_2O$ emissions caused by AOB and HET over the investigated operation period of 65 d (simulated values).

| (g $N_2O$-N/65 d) | AOB | HET | Total |
|---|---|---|---|
| SBR | 153.4 | 100.5 | 253.9 |
| Biofilm | 92.1 | 135.2 | 227.3 |

During the settling and decanting phases, high $N_2O$ concentrations could also be traced back to incomplete $NO_2$ denitrification. However, Figure 3 shows some differences between the measured and simulated concentrations. This could be attributed to the formation of a vertical $N_2O$ concentration profile after the stirrer was switched off due to the sinking of the biomass into the lower part of the reactor ($N_2O$ was measured in the upper part of the reactor). When the aeration and stirring were switched on, complete mixing was achieved again, and the $N_2O$ formed was evenly distributed in the reactor (confirmed by measurements). This effect is not represented in the model (CSTR), so a higher $N_2O$ concentration was computed.

The total emitted load was calculated on the basis of the off-gas measurements (Figure 4). Here, very similar results were obtained via measurement and simulation. In total, 253.9 g of $N_2O$-N was emitted during the investigated period of 65 days ($N_2O$ measurement period: measured 39.3 g of $N_2O$-N/8.94 d, simulated 39.6 g of $N_2O$-N/ 8.94 d), resulting in an emission factor of 15.3% of converted $NH_4$-N (see Table 2). Altogether, the results confirm that the digital model can adequately simulate the processes of $N_2O$ formation and emission as well as the nitrogen conversion of the investigated plant.

*4.2. Baseline Scenario Biofilm Reactor*

The dynamic influent data and operational settings of the investigated 65-day period were applied to the calibrated biofilm model. The modeling outcomes showed that due to high $O_2$ concentrations in the bulk phase and alkalinity-limited $NH_4$ conversion, aerobic conditions were established in all biofilm layers during the aerated phase (average biofilm thickness = 1.2 mm). The $O_2$ concentrations reached in the different biofilm layers (2.6–3.3 mg/L in the outermost biofilm layer, 0.3–2.6 mg/L in the innermost biofilm layer) depended on the load situation and the duration of the aerated phase (15 or 30 min, see Figure 5). Nevertheless, the AOB were mainly active in the first and second layers (together > 90% of the total AOB activity).

The different $O_2$ concentrations resulted in varying $N_2O$ formation factors over the biofilm layers; the lowest formation factors were achieved in the outermost biofilm layer. Here, a basic formation factor of approximately 7% was determined during the aerated phase, similar to that in the SBR. However, lower formation factors (2.8–7%) were achieved between days 0 and 5, 15 and 28, and 42 and 65 mainly due to a reduction in the AOB-specific $NH_4$-N conversion rate (see Figure 4). In the second biofilm layer, formation factors varied between 2.8 and 9.7%. In total, the $N_2O$ load formed by AOB was about 40% lower than that in the SBR (see Table 2).

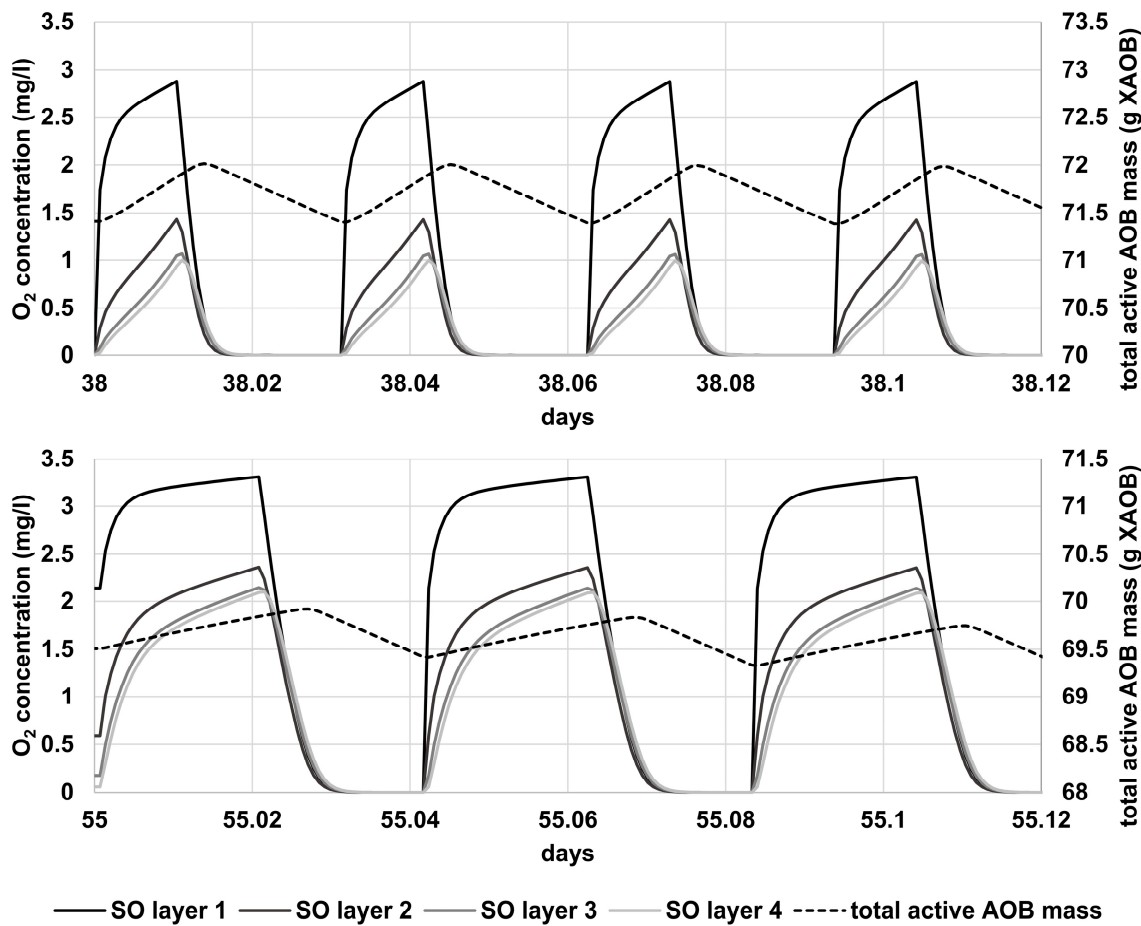

**Figure 5.** Oxygen profiles and total active AOB mass in the biofilm for two operation periods.

HET are responsible for 59.5% of total $N_2O$ emissions (see Table 2), implying that incomplete denitrification is the main $N_2O$ source. This is due to more heterotrophic biomass in the system compared with the SBR.

The total emissions were slightly (10%) lower in the biofilm system (227.3 g of $N_2O$-N were emitted during the investigated period of 65 days, resulting in an emission factor of 13.5% of converted $NH_4$-N). A significant decrease in $N_2O$ emissions was achieved between day 53 and day 65 due to the longer aeration interval (12 h/d) and the low influent load (see Figure 4).

As reported by [20], significantly higher $N_2O$ concentrations were observed in the biofilm than in the bulk phase when aeration was switched on. However, since there was no denitrification capacity, a reduction in $N_2O$ emissions could not be achieved. Instead, the $N_2O$ diffused into the bulk phase during the unaerated phase and was subsequently stripped at the beginning of the aerated phase.

In summary, $N_2O$ was formed not only by AOB but also by HET. In the biofilm reactor, denitrification was the main source. This implies that the impact on HET must also be considered when evaluating the measures investigated in the following scenarios. For the same N conversion, $N_2O$ formation by AOB was significantly lower in the biofilm system due to lower specific AOB activity.

### 4.3. S1: Effect of Different Aeration Cycles

Figure 6 shows the emissions caused by AOB and HET over the investigated 65-day period (baseline) as well as the average AOB-related $NH_4$-N conversion rate calculated for different aeration cycles. Moreover, the aerated time per day (h/d) is given. Due to the settling and decanting phases, a shorter aerated time per day resulted for the SBR reactor when the same aeration cycle was chosen.

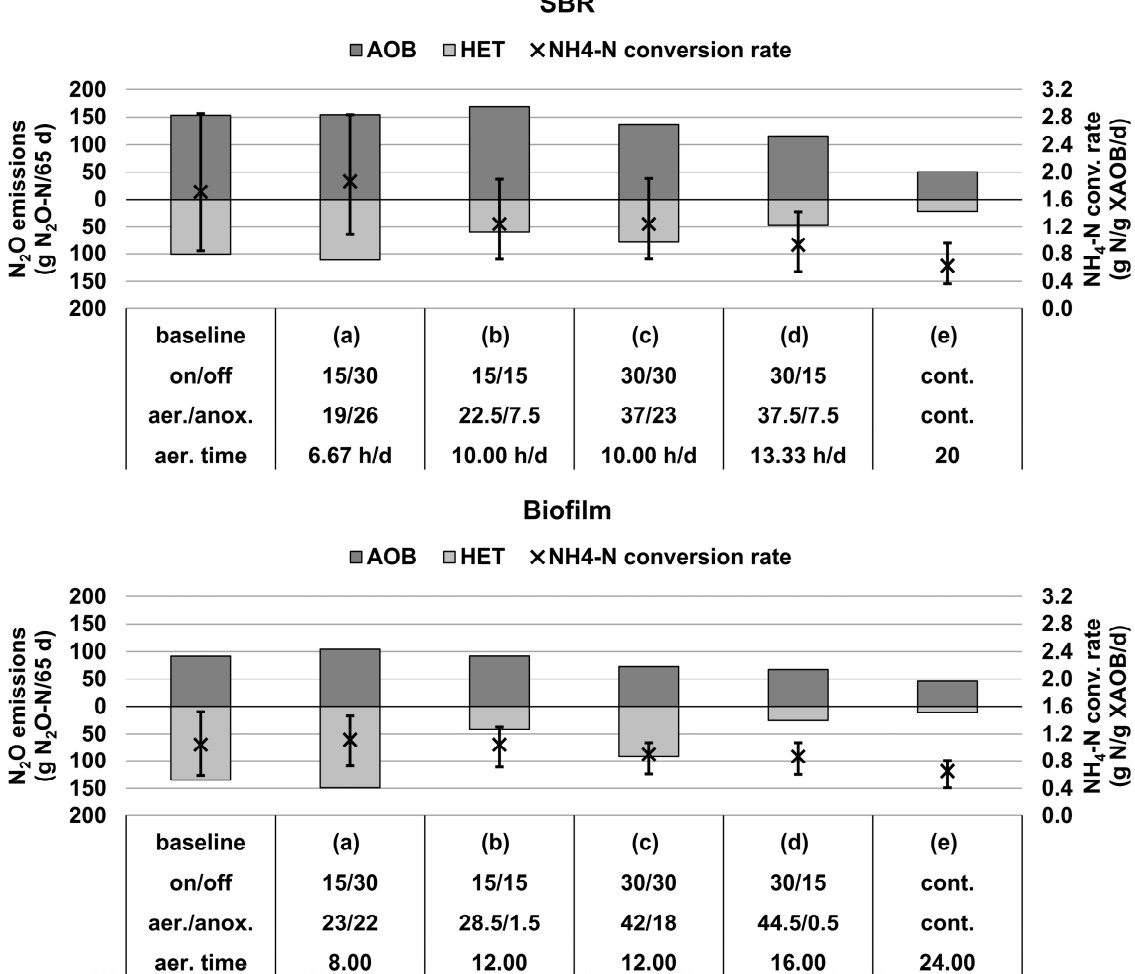

**Figure 6.** $N_2O$ emissions caused by AOB and HET (bars) over the investigated 65-day period and mean AOB-related $NH_4$-N conversion rate (error bars: min. and max. conversion rate) calculated for different aeration cycles.

Additionally, continuous aeration was tested. Since NOB suppression, in this case, could no longer be accomplished by alternating aerobic and anoxic intervals, NOB activity was manually set to zero to ensure comparable boundary conditions.

The results confirmed that the higher the aerated time per day, the lower the AOB-related $NH_4$-N conversion rate. Accordingly, since $N_2O$ formation is coupled with the AOB-related $NH_4$-N conversion rate, a strong decrease in $N_2O$ emissions caused by AOB could be observed for a longer aerated time per day (exception: "15/15" in SBR). Furthermore, as increased $N_2O$ formation by AOB was observed at the beginning of the anoxic phase, a low number of aeration cycles also has a positive effect (see aeration cycles "15/15" and "30/30" in Figure 6).

Higher fluctuations of the $NH_4$-N conversion rate were found in the SBR. This was related to a lower fluctuation of the total active biomass in the system. In the SBR reactor, the biomass was directly coupled to the influent (the influent contained microorganisms); in addition, biomass was withdrawn during the decanting phase. Both effects influenced the steady-state biomass concentration. As a consequence, the AOB-related $NH_4$-N conversion rate was mainly influenced by the aerated time per day (the converted $NH_4$-load and biomass concentration were constant). In the biofilm reactor, the fluctuations in the AOB-related $NH_4$-N conversion rate were much less pronounced and generally much lower than those in the SBR. Already in the baseline scenario, the conversion rates below the upper threshold for the determination of the $N_2O$ emission factor were calculated. This

was due to the possibility of maintaining a higher AOB mass, especially in high-load/stress situations (e.g., aeration cycle "15/30": 49.5 g of AOB in the SBR and 69.9 g of AOB in the biofilm reactor) for a short period. With an increased aerated time per day, the AOB mass declined (e.g., aeration cycle "30/15": 40 g of AOB in the biofilm reactor) because lower specific conversion rates were already sufficient for reaching the maximum $NH_4$-N conversion achievable under the given boundary conditions. A high AOB mass can only be maintained long-term if repeating high-load situations occur.

As expected, HET contributed more to $N_2O$ emissions as the length of the anoxic phase increased. Since aerobic conditions still prevailed at the beginning of the unaerated phase, longer unaerated phases were associated with longer anoxic phases and higher levels of $N_2O$ formation. Thus, higher levels of $N_2O$ formation were observed in the two scenarios with a 30 min unaerated phase. Moreover, a lower number of aeration cycles was associated with increased $N_2O$ formation by HET, compensating for the decreased $N_2O$ formation by AOB in the biofilm reactor.

Summarizing the results described, the selection of the aeration cycle had different effects regarding $N_2O$ formation:

- For a constant converted $NH_4$-N load, longer aerobic phases resulted in a reduced specific AOB activity because there was more time for aerobic metabolism.
- High $N_2O$ formation factors were calculated at the beginning of the unaerated phase due to low $O_2$ concentrations. For that reason, a smaller number of aeration cycles diminished $N_2O$ formation (AOB were less frequently exposed to low $O_2$ concentrations).
- A smaller number of aeration cycles increased the $N_2O$ formation by HET due to prolonged anoxic phases.

Overall, therefore, an optimum must be found between longer aeration intervals to activate additional AOB biomass, shorter anoxic phases to avoid $N_2O$ formation by HET, and a minimized number of aeration cycles to avoid environments with low $O_2$ concentrations. In addition, NOB suppression must continue to be ensured. In all scenarios, comparable N conversion was achieved because $NH_4$ conversion was limited by the alkalinity.

Since the focus of the practical application is on minimizing the total direct and indirect $CO_2$e emissions, the total emitted $CO_2$e load (not only the emitted $N_2O$ load) is the target parameter for plant control. For this reason, Figure 7 summarizes the estimated $CO_2$e emissions resulting from the release of $N_2O$ and the consumption of energy for aeration (other energy consumers are neglected since only the aeration settings were changed in this scenario). For the calculation of the $CO_2$e emissions from energy consumption, the recent emission factor of the German energy mix (375 g $CO_2$e/kWh) was employed. The results clearly showed that the $N_2O$ emissions avoided by the tested aeration strategies exceeded the emissions from the additional energy consumption by far (It must be emphasized that an experimental plant with high $N_2O$ emissions and relatively low energy consumption (good $O_2$ transfer) was examined here, so these results are not directly transferable to large-scale plants.).

Altogether, for both systems, the simulations resulted in the lowest emissions for the aeration cycle "30/15", not considering continuous aeration as an option. However, experience from pilot plant operations showed that sufficient suppression of NOB activity could not be achieved with a "30/10" aeration cycle, so problems can also be expected with the "30/15" aeration cycle. Therefore, an aeration cycle of "15/15" is recommended for the biofilm system, as this minimizes $N_2O$ formation by HET (the dominant $N_2O$ source). For the SBR, on the other hand, a "30/30" aeration cycle was selected as the preferred setting. The higher $N_2O$ production by HET can be compensated in this case by a lower $N_2O$ formation by AOB. In total, $N_2O$ emissions could be reduced by 15% (SBR) and 41% (biofilm) compared with the baseline scenarios. The results also confirmed that the mechanisms of $N_2O$ formation and conversion differed between the different systems, even though both systems were operated identically.

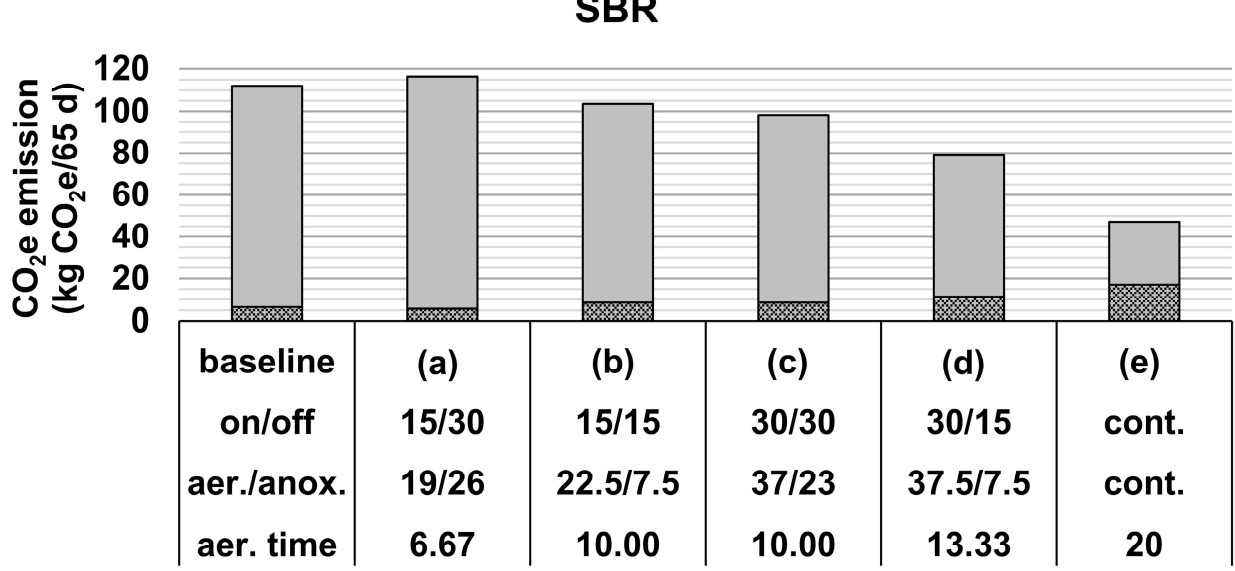

**SBR**

| baseline | (a) | (b) | (c) | (d) | (e) |
| --- | --- | --- | --- | --- | --- |
| on/off | 15/30 | 15/15 | 30/30 | 30/15 | cont. |
| aer./anox. | 19/26 | 22.5/7.5 | 37/23 | 37.5/7.5 | cont. |
| aer. time | 6.67 | 10.00 | 10.00 | 13.33 | 20 |

☒ energy aeration  ☐ N2O emissions

**Biofilm**

| baseline | (a) | (b) | (c) | (d) | (e) |
| --- | --- | --- | --- | --- | --- |
| on/off | 15/30 | 15/15 | 30/30 | 30/15 | cont. |
| aer./anox. | 23/22 | 28.5/1.5 | 42/18 | 44.5/0.5 | cont. |
| aer. time | 8.00 | 12.00 | 12.00 | 16.00 | 24.00 |

☒ energy aeration  ☐ N2O emissions

**Figure 7.** $CO_2e$ emissions resulting from the release of $N_2O$ and the consumption of energy for aeration.

### 4.4. S2: Effect of Different $O_2$ Concentrations

Figure 8 summarizes the modeling outcomes (emissions caused by AOB and HET over the investigated 65-day period and mean AOB-related $NH_4$-N conversion rate) for different target $O_2$ concentrations (2, 3, and 4 mg/L). A strong decrease in $NH_4$ conversion was observed at concentrations of <2 mg/L; therefore, lower concentrations were not investigated. With reference to the previous section, aeration cycles of "30/30" for the SBR and "15/15" for the biofilm reactor were selected.

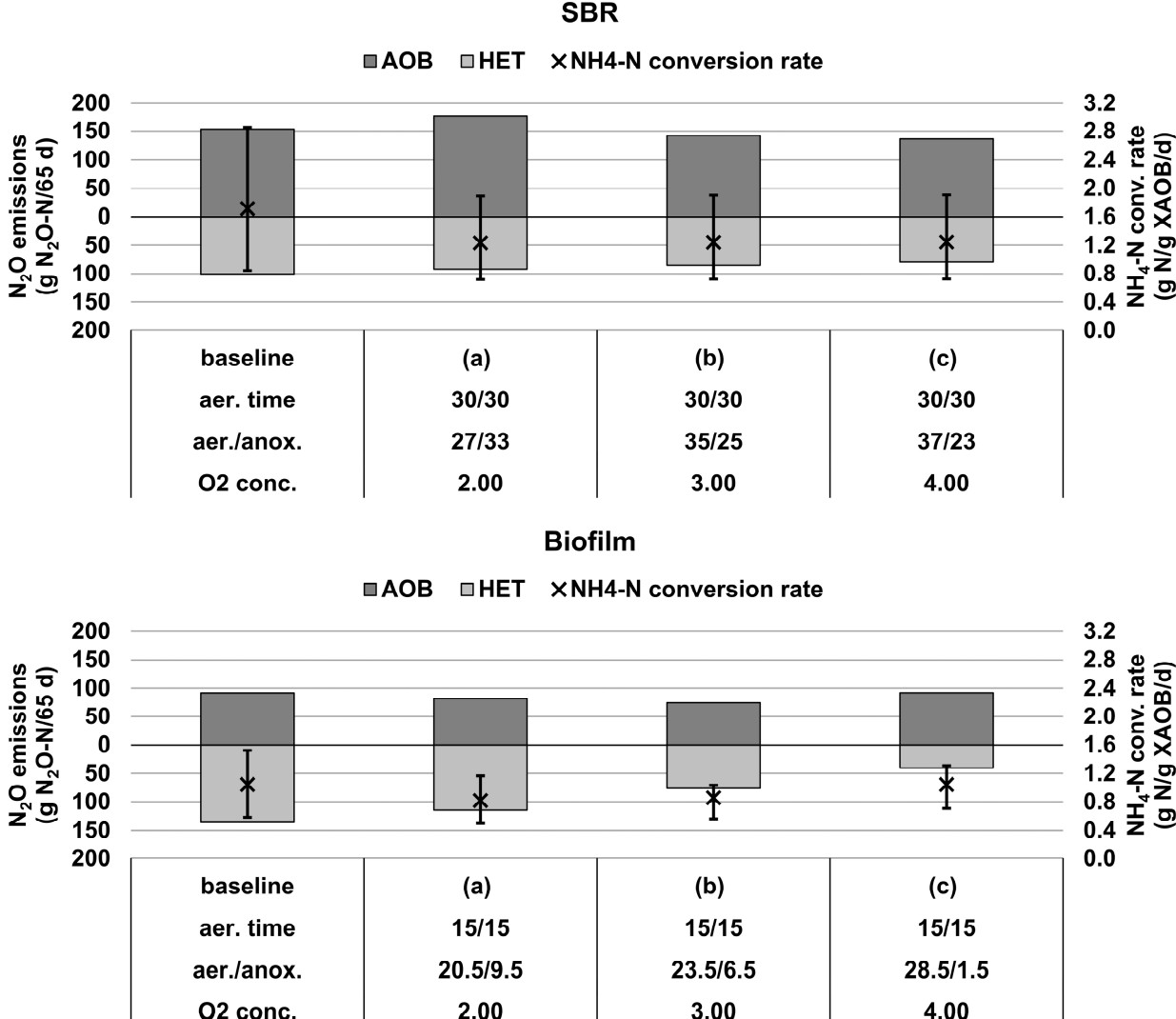

**Figure 8.** $N_2O$ emissions caused by AOB and HET (bars) over the investigated 65-day period as well as the mean AOB-related $NH_4$-N conversion rate (error bars: min. and max. conversion rate) calculated for different $O_2$ concentrations.

For both reactors, a decrease in total $N_2O$ emissions was observed for higher $O_2$ concentrations. In the SBR, higher $O_2$ concentrations were associated with a moderate reduction in $N_2O$ formation by both AOB and HET. While the former was related to reduced $O_2$ concentration-dependent $N_2O$ formation factors, the latter was due to slightly shorter anoxic phases (resulting from the higher $O_2$ mass in the reactor when aeration is switched off). The $N_2O$ emissions from the biofilm reactor were more strongly affected by the $O_2$ concentration. As with the SBR, a decrease in $N_2O$ formation by HET was observed with an increase in $O_2$ concentration, which was also due to shorter anoxic phases. Even at an $O_2$ concentration of 2 mg/L in the bulk phase, an aerobic environment ($O_2$ concentration > 0.2 mg/L) was still achieved in the innermost biofilm layer during the aerated phase. At lower bulk $O_2$ concentrations, however, slightly lower AOB-induced $N_2O$ emissions were calculated. This could be attributed to several effects:

- At an $O_2$ concentration of 4 mg/L, a lower AOB mass was established in the system because more time was available for aerobic N conversion due to the longer aerobic phases (approx. 7 min more per aeration cycle at 4 mg/L than at 2 mg/L → lower conversion rates were already sufficient). Excess biomass decayed over time (see also

next section). At a constant N conversion, the specific AOB conversion rate increased as a result (especially during N load peaks).

- While N conversion at 4 mg/L occurred in the two outer layers, N conversion at 2 mg/L was mostly localized in the outermost layer. The determined formation factors in the outermost layer at 2 mg/L and the two outermost layers at 4 mg/L were relatively similar during the aerated phase. However, as the $O_2$ concentration in the biofilm dropped after aeration was turned off, an increase in the formation factors was observed. Since the $O_2$ concentration in the biofilm fell more slowly at 4 mg/L, aerobic N conversion could occur for a longer period in a low-$O_2$ environment, inducing higher $N_2O$ emissions.

Summarizing the results, $O_2$ concentrations of $\geq 3$ mg/L only have a minor influence on $N_2O$ emissions of the SBR. In the biofilm reactor, a decrease in biomass and a longer exposure to low $O_2$ concentrations were associated with slightly higher AOB-related $N_2O$ emissions at higher $O_2$ concentrations. However, a strong reduction in heterotrophic $N_2O$ formation was observed at higher $O_2$ concentrations, resulting in a net decrease in the total emissions.

### 4.5. S3: Effect of Different Biomass Contents

In the SBR, higher TSS concentrations due to improved settling were associated with higher AOB activity and reduced autotrophic $N_2O$ formation. However, higher TSS concentrations promoted heterotrophic $N_2O$ formation, so the total emissions did not significantly change.

In the biofilm, the simulations ("15/15" aeration cycle, $O_2$ concentration = 4 mg/L) with different erosion velocities resulting in different biofilm thicknesses showed that the total biomass in the reactor does not depend on the biofilm thickness as soon as a steady state is reached (Figure 9 right). Similar to the studies on different $O_2$ concentrations, only the biomass required for the conversion processes (the N conversion is identical here in all scenarios) was retained; the excess biomass decayed. Consequently, the $N_2O$ emissions caused by AOB were similar for the investigated biofilm thicknesses. The total $N_2O$ emissions of the thicker biofilm were indeed slightly higher due to a faster decrease in the $O_2$ concentration (more $O_2$-consuming microorganisms in the outer biofilm layer), resulting in a longer anoxic phase and a slightly higher activity of HET (Figure 9 left).

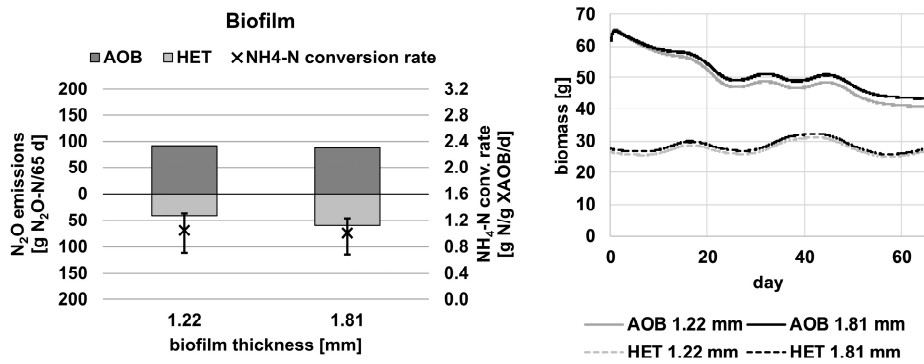

**Figure 9.** $N_2O$ emissions caused by AOB and HET (bars) over the investigated 65-day period as well as the mean AOB-related $NH_4$-N conversion rate (error bars: min. and max. conversion rate) calculated for different biofilm thicknesses (**left**) and total mass of AOB and HET in the reactor (**right**).

It must be emphasized that the simulations started from a steady state, which was reached after more than 500 days. In reality, the conservation of biomass can definitely be achieved over shorter periods. Some studies have confirmed that AOB can be reactivated after longer periods without substrate availability (e.g., [25]). Applying the model, a higher biomass concentration could be maintained by repeated high-load phases. For example, the alkalinity of the inflow could be increased for one day every 7 days. However, since

this resulted in increased N conversion on the one hand and higher $N_2O$ emissions on the other hand, this was not further examined in this work.

### 4.6. S4: Equalization of the Influent (Only Biofilm Reactor)

Equalizing the inflow did not significantly affect the total $N_2O$ emissions of the biofilm reactor, as shown in Figure 10, although peaks in the inflowing $NH_4$-N load could be reduced. Emissions were just shifted from previously higher-loaded phases to previously lower-loaded phases. In the literature, increased $N_2O$ formation has been reported as a result of shock loads (e.g., [26,27]). However, this effect did not play a role in this study because the load change always occurred over several days.

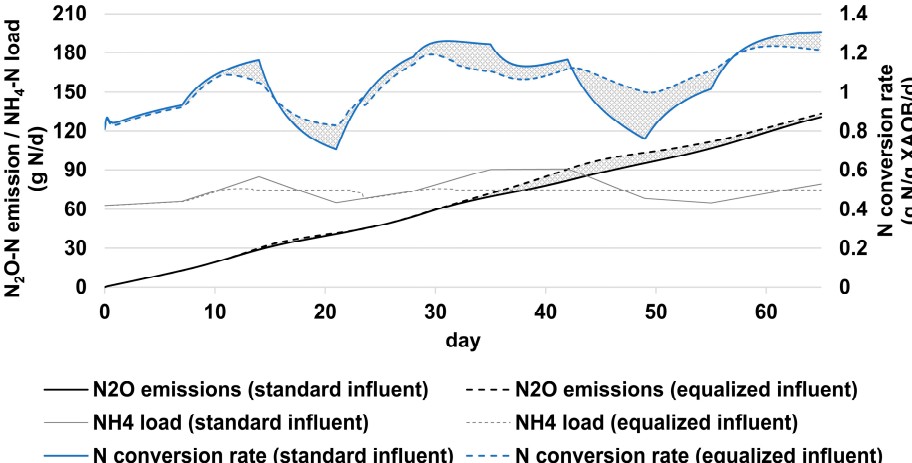

**Figure 10.** Total cumulated $N_2O$ emissions, inflowing $NH_4$ load, and N conversion rate for standard and equalized inflow.

## 5. Summary and Conclusions

The results of this study confirmed that there are substantial differences in $N_2O$ formation between biofilm systems and systems with suspended biomass, even though both systems are operated identically. Therefore, strategies to reduce emissions are not directly transferable. A combination of measurement and modeling, however, can support the identification of the causes of high emissions and the development of appropriate mitigation measures for both systems. Since measurements inside biofilms are usually difficult and expensive, calibrated mathematical models are essential to detect and evaluate the environmental conditions of these systems (model calibration via measurements in the bulk phase) [20].

It was proven in this study that reducing the specific AOB activity can significantly decrease $N_2O$ formation without affecting the reactor performance. The relationship between $N_2O$ formation and AOB activity has been known for years, but until now, AOB activity has not been used as a target parameter for a control strategy. With the newly developed approach, a main cause of high autotrophic $N_2O$ formation can directly be addressed. The described concept is a step from the measurement-based plant control toward a model-based control employing digital twins. For the boundary conditions investigated in this study, the biofilm system proved to be more advantageous, as a higher biomass content could be maintained in the biofilm. Thus, a conversion rate below the upper threshold according to Figure 2 was achieved for more than 60% of the examined period, resulting in the $N_2O$ formation by AOB being approx. 40% lower than that in the SBR.

The highest reduction in autotrophic $N_2O$ formation was achieved by changing the aeration cycles (SBR: 25%; Biofilm: 27%). A longer aerated time per day resulted in reduced $N_2O$ formation, as there was more time for nitrogen conversion (leading to a lower specific AOB activity). In addition, a lower number of aeration cycles had a positive effect on au-

totrophic $N_2O$ formation, as $O_2$ deficiency situations (after aeration is turned off) occurred less frequently. An essential function of intermittent aeration in pure nitritation systems is, however, the prevention of $NO_2$ oxidation by NOB, which must also be considered when selecting the aeration settings. The pilot plant operation showed that with the aeration cycles "15/15" and "30/30", the suppression of $NO_2$ formation can be ensured in the long term. Nevertheless, longer aerated phases up to continuous aeration can be selected for short periods, for example, to avoid increased $N_2O$ formation during load peaks.

For the systems investigated, an $O_2$ concentration of $\geq 3$ mg/L in the bulk phase proved to be ideal. Higher concentrations did not lead to a further reduction in $N_2O$ formation. The biofilm thickness and the TSS concentration played only a subordinate role when considering a steady state since in the long term, only the system-specific biomass, depending on the converted nitrogen load and the shear rate/sludge removal rate, is maintained. Temporary storage and equalization of the inflow had no effect on the $N_2O$ emissions over the entire investigated period since the emissions were merely shifted (short-term load peaks were not considered).

A disadvantage of the biofilm system is the higher risk of $N_2O$ formation by HET (here 34.5% more than in the SBR) due to the higher activity of HET establishing in the biofilm reactor (depending on the biofilm thickness and substrate diffusion). $N_2O$ formation by denitrification is unavoidable in both systems under the given boundary conditions since the $N_2O$ reduction is inhibited by $HNO_2$ ($NO_2$ concentration: 200–300 mg N/L; pH value: 6.0–6.5). It can be stated that for biofilm systems, incomplete denitrification is the main $N_2O$ source. With regard to heterotrophic $N_2O$ formation, the longest possible aerobic time per day should be aimed for. Moreover, a higher number of shorter aeration cycles is advantageous.

In summary, it was proven that $N_2O$ emissions can be reduced with appropriate aeration strategies by maintaining a low AOB activity (biofilm system: reduction in emissions by 34%). For the biofilm system, however, the measures investigated by [20] for the implementation of heterotrophic denitrification not as a source but as an $N_2O$ sink showed higher effectiveness (reduction in emissions by 75%). Therefore, the main focus should be on a reduction in the $HNO_2$ concentration by an adapted operation strategy (e.g., single-stage deammonification instead of two-stage deammonification, resulting in lower $NO_2$ accumulation). Moreover, sufficient anoxic environments must be provided (zoning in biofilm or intermittent aeration). That way, additional $N_2O$ formed by AOB can be degraded by denitrification, resulting in lower net emissions. $N_2O$ formation by AOB should be addressed after $N_2O$ denitrification has been successfully established.

Due to the high complexity and a large number of simultaneously occurring processes, a combination of deterministic models with intelligent optimization algorithms offers a high potential for minimizing greenhouse gas emissions in wastewater treatment. Thus, the control strategy can be automatically adapted to a specific operating situation, addressing autotrophic and/or heterotrophic $N_2O$ formation. Furthermore, denitrification capacities could be flexibly adjusted to the load situation.

**Author Contributions:** Conceptualization and methodology, M.B. and A.F.; investigation and modeling, A.F.; writing and visualization, A.F.; supervision and project administration, M.B. All authors have read and agreed to the published version of the manuscript.

**Funding:** This work was carried out as part of the MiNzE project ('Minimization of the $CO_2$ footprint by adapted process development in process water treatment—testing of the MiNzE process in an immersed fixed bed', FKZ: 02WQ1482B). We thank the German Federal Ministry of Education and Research for financial support.

**Data Availability Statement:** The data presented in this study are openly available at https://doi.org/10.25835/psv2gfsu (accessed on 30 May 2023).

**Conflicts of Interest:** The authors declare no conflict of interest. The funders had no role in the design of the study; in the collection, analyses, or interpretation of data; in the writing of the manuscript; or in the decision to publish the results.

# Appendix A

| | $SS$ | $SO$ | $SNH4$ | $SNO3$ | $SNO2$ | $SN2O$ | $SN2O_{Gas}$ | $SN2O_{emittiert}$ | $SN2$ | $SALK$ | $XS$ | $XI$ | $SI$ | $XND$ | $SND$ | $XH$ | $XAOB$ | $XAOBI$ | $XNOB$ | $XNOBI$ | $XAN$ | $nBF$ |
|---|---|---|---|---|---|---|---|---|---|---|---|---|---|---|---|---|---|---|---|---|---|---|
| **Gro_XH_NO3** | $\frac{-1}{Y_{HET,SS,NO3}}$ | 0 | $-1\,iNBM$ | $\frac{-1}{Y_{NO3}}$ | $\frac{1}{Y_{NO3}}$ | 0 | 0 | 0 | 0 | 0.009 | 0 | 0 | 0 | 0 | 0 | 1 | 0 | 0 | 0 | 0 | 0 | 0 |
| **Gro_XH_NO2** | $\frac{-1}{Y_{HET,SS,NO2}}$ | 0 | $-1\,iNBM$ | 0 | $\frac{-1}{Y_{NO2}}$ | $\frac{1}{Y_{NO2}}$ | 0 | 0 | 0 | 0.009 | 0 | 0 | 0 | 0 | 0 | 1 | 0 | 0 | 0 | 0 | 0 | 0 |
| **Gro_XH_N2O** | $\frac{-1}{Y_{HET,SS,N2O}}$ | 0 | $-1\,iNBM$ | 0 | 0 | $\frac{-1}{Y_{N2O}}$ | 0 | 0 | $\frac{1}{Y_{N2O}}$ | 0.009 | 0 | 0 | 0 | 0 | 0 | 1 | 0 | 0 | 0 | 0 | 0 | 0 |
| **Gro_XH_O2** | $\frac{-1}{Y_{HET,SS,O2}}$ | $1-\frac{1}{Y_{HET,SS,O2}}$ | $-1\,iNBM$ | 0 | 0 | 0 | 0 | 0 | 0 | $-0.005$ | 0 | 0 | 0 | 0 | 0 | 1 | 0 | 0 | 0 | 0 | 0 | 0 |
| **Dec_XH** | 0 | 0 | 0 | 0 | 0 | 0 | 0 | 0 | 0 | 0.003 | $1-fXI$ | $fXI$ | 0 | 0.0818 | 0 | $-1$ | 0 | 0 | 0 | 0 | 0 | 0 |
| **Gro_XAOB** | 0 | $\frac{\frac{48}{14}}{Y_{AOB}}+1$ | $\frac{-1}{Y_{AOB}}-iNBM$ | 0 | $\frac{1}{Y_{AOB}}\left(1-BF_{ges}\right)$ | $\frac{1}{Y_{AOB}}BF_{ges}$ | 0 | 0 | 0 | $-0.683$ | 0 | 0 | 0 | 0 | 0 | 0 | 1 | 0 | 0 | 0 | 0 | 0 |
| **Dec_XAOB** | 0 | 0 | 0 | 0 | 0 | 0 | 0 | 0 | 0 | 0.003 | $1-fXI$ | $fXI$ | 0 | 0.0818 | 0 | 0 | $-1$ | 0 | 0 | 0 | 0 | 0 |
| **Dec_XAOBI** | 0 | 0 | 0 | 0 | 0 | 0 | 0 | 0 | 0 | 0.003 | $1-fXI$ | $fXI$ | 0 | 0.0818 | 0 | 0 | 0 | $-1$ | 0 | 0 | 0 | 0 |
| **Gro_XNOB** | 0 | $\frac{\frac{-16}{14}}{Y_{NOB}}+1$ | $-1\,iNBM$ | $\frac{1}{Y_{NOB}}$ | $\frac{-1}{Y_{NOB}}$ | 0 | 0 | 0 | 0 | 0.005 | 0 | 0 | 0 | 0 | 0 | 0 | 0 | 0 | 1 | 0 | 0 | 0 |
| **Dec_XNOB** | 0 | 0 | 0 | 0 | 0 | 0 | 0 | 0 | 0 | 0.003 | $1-fXI$ | $fXI$ | 0 | 0.0818 | 0 | 0 | 0 | 0 | $-1$ | 0 | 0 | 0 |
| **Dec_XNOBI** | 0 | 0 | 0 | 0 | 0 | 0 | 0 | 0 | 0 | 0.003 | $1-fXI$ | $fXI$ | 0 | 0.0818 | 0 | 0 | 0 | 0 | 0 | $-1$ | 0 | 0 |
| **Gro_XAN** | 0 | 0 | $\frac{-1}{Y_{AN}}-iNBM$ | $\frac{14}{16}$ | $\frac{-1}{Y_{AN}}-\frac{14}{16}$ | 0 | 0 | 0 | $\frac{2}{Y_{AN}}$ | $-0.005$ | 0 | 0 | 0 | 0 | 0 | 0 | 0 | 0 | 0 | 0 | 1 | 0 |
| **Dec_XAN** | 0 | 0 | 0 | 0 | 0 | 0 | 0 | 0 | 0 | 0.003 | $1-fXI$ | $fXI$ | 0 | 0.0818 | 0 | 0 | 0 | 0 | 0 | 0 | $-1$ | 0 |
| **Hydrolyse_XS** | $1-fSI$ | 0 | 0 | 0 | 0 | 0 | 0 | 0 | 0 | 0 | $-1$ | 0 | $fSI$ | 0 | 0 | 0 | 0 | 0 | 0 | 0 | 0 | 0 |
| **Hydrolyse_XND** | 0 | 0 | 0 | 0 | 0 | 0 | 0 | 0 | 0 | 0 | 0 | 0 | 0 | $-1$ | 1 | 0 | 0 | 0 | 0 | 0 | 0 | 0 |
| **Ammonifikation_SND** | 0 | 0 | 1 | 0 | 0 | 0 | 0 | 0 | 0 | 0 | 0 | 0 | 0 | 0 | $-1$ | 0 | 0 | 0 | 0 | 0 | 0 | 0 |
| **Ac_AOB** | 0 | 0 | 0 | 0 | 0 | 0 | 0 | 0 | 0 | 0 | 0 | 0 | 0 | 0 | 0 | 0 | 1 | $-1$ | 0 | 0 | 0 | 0 |
| **De_AOB** | 0 | 0 | 0 | 0 | 0 | 0 | 0 | 0 | 0 | 0 | 0 | 0 | 0 | 0 | 0 | 0 | $-1$ | 1 | 0 | 0 | 0 | 0 |
| **Ac_NOB** | 0 | 0 | 0 | 0 | 0 | 0 | 0 | 0 | 0 | 0 | 0 | 0 | 0 | 0 | 0 | 0 | 0 | 0 | 1 | $-1$ | 0 | 0 |
| **De_NOB** | 0 | 0 | 0 | 0 | 0 | 0 | 0 | 0 | 0 | 0 | 0 | 0 | 0 | 0 | 0 | 0 | 0 | 0 | $-1$ | 1 | 0 | 0 |
| **Belueftung** | 0 | 1 | 0 | 0 | 0 | 0 | 0 | 0 | 0 | 0 | 0 | 0 | 0 | 0 | 0 | 0 | 0 | 0 | 0 | 0 | 0 | 0 |
| **N2O_Gastransfer** | 0 | 0 | 0 | 0 | 0 | $-1\,nBF$ | $1\,nBF$ | 0 | 0 | 0 | 0 | 0 | 0 | 0 | 0 | 0 | 0 | 0 | 0 | 0 | 0 | 0 |
| **N2O_Emission** | 0 | 0 | 0 | 0 | 0 | 0 | $-1\,nBF$ | $1\,nBF$ | 0 | 0 | 0 | 0 | 0 | 0 | 0 | 0 | 0 | 0 | 0 | 0 | 0 | 0 |
| **SN2O_gas_Rueckfuehrung** | 0 | 0 | 0 | 0 | 0 | 0 | $1\,nBF$ | 0 | 0 | 0 | 0 | 0 | 0 | 0 | 0 | 0 | 0 | 0 | 0 | 0 | 0 | 0 |

**Figure 1.** ASM matrix.

**Gro_XH_NO3**　$\mu_{\text{HET}}\,\eta_{\text{HET,anox}}\,f_{\text{T,XH}}\,f_{\text{NO3}}\,\dfrac{SS}{SS+K_{\text{HET,SS,NO3}}}\,\dfrac{K_{\text{HET,O2}}}{K_{\text{HET,O2}}+SO}\,\dfrac{SNH4}{SNH4+K_{\text{HET,NH4}}}\,\dfrac{SNO3}{SNO3+K_{\text{HET,NO3}}}\,XH$

**Gro_XH_NO2**　$\mu_{\text{HET}}\,\eta_{\text{HET,anox}}\,f_{\text{T,XH}}\,f_{\text{NO2}}\,\dfrac{SS}{SS+K_{\text{HET,SS,NO2}}}\,\dfrac{K_{\text{HET,O2}}}{K_{\text{HET,O2}}+SO}\,\dfrac{SNH4}{SNH4+K_{\text{HET,NH4}}}\,\dfrac{SNO2}{SNO2+K_{\text{HET,NO2}}}\,XH$

**Gro_XH_N2O**　$\mu_{\text{HET}}\,\eta_{\text{HET,anox}}\,f_{\text{T,XH}}\,f_{\text{N2O}}\,\dfrac{SS}{SS+K_{\text{HET,SS,N2O}}}\,\dfrac{K_{\text{HET,O2,N2O}}}{K_{\text{HET,O2,N2O}}+SO}\,\dfrac{SNH4}{SNH4+K_{\text{HET,NH4}}}\,\dfrac{SN2O}{SN2O+K_{\text{HET,N2O}}}\,\dfrac{K_{\text{HET,I,HNO2}}}{K_{\text{HET,I,HNO2}}+SHNO2}\,XH$

**Gro_XH_O2**　$\mu_{\text{HET}}\,f_{\text{T,XH}}\,\dfrac{SS}{SS+K_{\text{HET,SS,O2}}}\,\dfrac{SO}{K_{\text{HET,O2}}+SO}\,\dfrac{SALK}{SALK+K_{\text{HET,ALK}}}\,\dfrac{SNH4}{SNH4+K_{\text{HET,NH4}}}\,XH$

**Dec_XH**　$b_{\text{HET}}\,f_{\text{T,XH}}\,XH$

**Gro_XAOB**　$\mu_{\text{AOB}}\,f_{\text{T,XA}}\,\dfrac{SO}{K_{\text{AOB,O2}}+SO}\,\dfrac{K_{\text{AOB,I,NH3}}}{K_{\text{AOB,I,NH3}}+SNH3}\,\dfrac{K_{\text{AOB,I,HNO2}}}{K_{\text{AOB,I,HNO2}}+SHNO2}\,\dfrac{SNH4}{K_{\text{AOB,NH4}}+SNH4}\,\dfrac{SALK}{K_{\text{AOB,ALK}}+SALK}\,XAOB$

**Dec_XAOB**　$\left(b_{\text{AOB}}\,f_{\text{T,XA}}\,\dfrac{SO}{K_{\text{AOB,O2}}+SO}+\eta_{\text{AOB}}\,b_{\text{AOB}}\,f_{\text{T,XA}}\,\dfrac{K_{\text{AOB,O2}}}{K_{\text{AOB,O2}}+SO}\right)XAOB$

**Dec_XAOBI**　$\left(b_{\text{AOB}}\,f_{\text{T,XA}}\,\dfrac{SO}{K_{\text{AOB,O2}}+SO}+\eta_{\text{AOB}}\,b_{\text{AOB}}\,f_{\text{T,XA}}\,\dfrac{K_{\text{AOB,O2}}}{K_{\text{AOB,O2}}+SO}\right)XAOBI$

**Gro_XNOB**　$\mu_{\text{NOB}}\,f_{\text{T,XA}}\,\dfrac{SO}{K_{\text{NOB,O2}}+SO}\,\dfrac{K_{\text{NOB,I,NH3}}}{K_{\text{NOB,I,NH3}}+SNH3}\,\dfrac{K_{\text{NOB,I,HNO2}}}{K_{\text{NOB,I,HNO2}}+SHNO2}\,\dfrac{SNO2}{K_{\text{NOB,NO2}}+SNO2}\,XNOB$

**Dec_XNOB**　$\left(b_{\text{NOB}}\,f_{\text{T,XA}}\,\dfrac{SO}{K_{\text{NOB,O2}}+SO}+\eta_{\text{NOB}}\,b_{\text{NOB}}\,f_{\text{T,XA}}\,\dfrac{K_{\text{NOB,O2}}}{K_{\text{NOB,O2}}+SO}\right)XNOB$

**Dec_XNOBI**　$\left(b_{\text{NOB}}\,f_{\text{T,XA}}\,\dfrac{SO}{K_{\text{NOB,O2}}+SO}+\eta_{\text{NOB}}\,b_{\text{NOB}}\,f_{\text{T,XA}}\,\dfrac{K_{\text{NOB,O2}}}{K_{\text{NOB,O2}}+SO}\right)XNOBI$

**Gro_XAN**　$\mu_{\text{AN}}\,f_{\text{T,AN}}\,\dfrac{K_{\text{AN,I,O2}}}{K_{\text{AN,I,O2}}+SO}\,\dfrac{SNH4}{K_{\text{AN,NH4}}+SNH4}\,\dfrac{SNO2}{K_{\text{AN,NO2}}+SNO2+\dfrac{SNO2^{2}}{K_{\text{AN,I,NO2}}}}\,XAN$

**Dec_XAN**　$\left(b_{\text{AN}}\,f_{\text{T,AN}}\,\dfrac{SO}{K_{\text{AN,I,O2}}+SO}+\eta_{\text{AN}}\,b_{\text{AN}}\,f_{\text{T,AN}}\,\dfrac{K_{\text{AN,I,O2}}}{K_{\text{AN,I,O2}}+SO}\right)XAN$

**Hydrolyse_XS**　$kH20\,f_{\text{T,Hyd}}\,\dfrac{XS}{\min\,(XH,\,0.001)}\,\dfrac{1}{f_{\text{T,Hyd}}\,KX+\dfrac{XS}{\min\,(XH,\,0.001)}}\left(\dfrac{SO}{K_{\text{HET,O2}}+SO}+v_{\text{anox}}\,\dfrac{K_{\text{HET,O2}}}{K_{\text{HET,O2}}+SO}\,\dfrac{SNO3+SNO2+SN2O}{KNOx+SNO3+SNO2+SN2O}+v_{\text{an}}\,\dfrac{K_{\text{HET,O2}}}{K_{\text{HET,O2}}+SO+SNO3+SNO2+SN2O}\right)XH$

**Hydrolyse_XND**　$Hydrolyse_{\text{XS}}\,\dfrac{XND}{\min\,(XS,\,0.001)}$

**Ammonifikation_SND**　$kA20\,f_{\text{T,XH}}\,SND\,XH$

**Ac_AOB**　$x_{\text{Ak,AOB}}\,XAOBI$

**De_AOB**　$x_{\text{De,AOB}}\,XAOB$

**Ac_NOB**　$x_{\text{Ak,NOB}}\,XNOBI$

**De_NOB**　$x_{\text{De,NOB}}\,XNOB$

**Belueftung**　$akla20\,(SO_{\text{sat}}-SO)\,1.024^{\,(T-20)}$

**N2O_Gastransfer**　$r_{\text{N2O,Gastransferrate}}$

**N2O_Emission**　$r_{\text{N2O,Emissionsrate}}$

**SN2O_gas_Rueckfuehrung**　$Q_{\text{RF}}$

**Figure 2.** ASM equations.

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
