# Peer review of "Activity of Ammonium-Oxidizing Bacteria—An Essential Parameter for Model-Based N2O Mitigation Control Strategies for Biofilms"

_water, doi:10.3390/w15132389_

Round 1

Reviewer 1 Report

 The authors must underline the major findings of their work and explain how the use of their proposed procedures and materials represent a progress to other similar studies. The novelty must be better pointed. In conclusions the attitude to the goal is necessary. 

Some general remarks:

1. The title is proper.

2. The layout is clear and content is logically divided.

3. Test methods are proper and exactly described.

4. References are up – to date.

Detail remarks:

Please respect author guide.

Author Response

Dear reviewer,

Thank you very much for the feedback. We agree that the main findings have not been described clearly enough. We have revised the chapters 3 and 5 to highlight more clearly the newly developed approach of using AOB activity as a target parameter for low-emission plant control, moving from measurement-based plant control towards model-based plant control via digital twins. The results of this study can be used in future for the development of new model-based N2O mitigation strategies.

  • ll. 209 – 219: According to [23], N2O formation by AOB is dynamically computed as a fraction of converted NH4, relying on N2O formation factors calculated by summing up the influ-ences of the volumetric NH4-N conversion rate [g N/m³/d] and the concentrations of NO2-N and O2. The volumetric NH4-N conversion rate is used as an indicator for the AOB activity. However, since the specific AOB activity does not depend on the reactor volume but on the ratio of nitrogen load and converting biomass [g NH4-N/g AOBactive], the abso-lute mass of active AOB in the reactor or the biofilm section evaluated must be considered (e.g. [7, 8]). It was decided not to select a VSS-related conversion rate because this parame-ter is mainly influenced by the growth of heterotrophic microorganisms. Instead, a new N2O formation factor considering the specific AOB activity was introduced to the model.
  • ll. 232 -234: The new approach can be used to simulate the effects of a variable AOB activity on N2O formation and to implement model-based control algorithms to minimize N2O emissions.
  • ll. 521 – 526: The relationship between N2O formation and AOB activity has been known for years, but until now AOB activity has not been used as a target parameter for a control strategy. With the newly developed approach, causes of high autotrophic N2O formation can directly be addressed. The described concept is a step from the measurement-based plant control to-wards a model-based control employing digital twins.

Reviewer 2 Report

The paper deals with a very interesting subject and will certainly find a scientific echo. It is recommended not to use abbreviations in the title of the work. More precisely, it is AOB. Therefore, it is recommended to write the full name in nalsov. Definitions should be removed in the methods chapter. Although they refer to the methodology, it is better to write them in the introductory part of the chapter. Also, when describing the methods used, it should be stated exactly what was done and avoid the term that says a particular method was done according to a literature source, e.g., number 20:
Measurement data from the pilot plant described in [20] was used for this modeling study. Detailed information about this plant can be found in [20].
And: an extended ASM model developed by [23] and adapted for biofilm modeling by 146 [20] was used (provided here: [24]). The ASM model includes the following biological 147 and physical processes:
Figure 1: Inlet concentrations of NH4-N and alkalinity, and aeration mode over the operating period studied should be 146 moved to the introductory part of the discussion because the methodology is not the same.
Thus, this part of the methodology needs to be significantly improved.

The paper deals with a very interesting subject and will certainly find a scientific echo but it shuld improve in meaterial and methosd part. 

Author Response

Dear reviewer,

Thank you for the valuable comments. These were very helpful for us.

  • We have removed the abbreviation “AOB” from the title.
  • We added a reference to the definitions regarding aeration in the introduction (ll. 94-96). However, since these definitions are not officially used, we decided to leave them in the chapter "Material and Methods" to avoid confusion.
  • Since the main characteristics of the pilot plant have already been summarized in the text, we have removed the reference to source 20 when describing the pilot plant. We have also added information on the calculation of autotrophic N2O formation (ll. 158-160). Moreover, we additionally provide an overview on the ASM matrix in Appendix A. However, since the ASM model used was not developed as part of this study, we think that references to the sources cited are necessary. It would go beyond the scope of this study to explain all extensions of the complex model made in the past.
  • The information shown in Figure 1 serve as a static input for the modeling (comparable e. g. to the reactor volume). They are neither simulation results nor changed during the simulations. For that reason, we decided to leave the figure in chapter 2. However, we have modified caption to clarify this (ll. 150-151).

Reviewer 3 Report

Dear author

I believe that your manuscript is well written. However, the figures should be improved (all them), as the titles and scale numbers are too small in most of them. I don't like the way you exposed the content in the tables. But, it was your option!

I suppose it is a good enough manuscript to be published on the Water review.

Good lucky,

the reviewer

Author Response

Dear reviewer,

thank you for the very positive comment. We have changed the font size of the figures so that they are now better readable. We also improved the design of the tables.